# Characterization of a composite with enhanced attraction to savannah tsetse flies from constituents or analogues of tsetse refractory waterbuck (*Kobus defassa*) body odor

Benson M. Wachira[1,2]*, Joy M. Kabaka[1], Paul O. Mireji[2,3], Sylvance O. Okoth[2], Margaret M. Nganga[1], Robert Changasi[2], Patrick Obore[2], Bernard Ochieng'[4], Grace A. Murilla[2], Ahmed Hassanali[1]

**1** Kenyatta University, Nairobi, Kenya, **2** Biotechnology Research Institute—Kenya Agricultural and Livestock Research Organization, Kikuyu, Kenya, **3** Kenya Medical Research Institute, Kilifi, Kenya, **4** Shimba Hills National Reserve—Kenya Wildlife Service, Kwale, Kenya

* wachirabenson3@gmail.com

## Abstract

Savannah tsetse flies avoid flying toward tsetse fly-refractory waterbuck (*Kobus defassa*) mediated by a repellent blend of volatile compounds in their body odor comprised of δ-octalactone, geranyl acetone, phenols (guaiacol and carvacrol), and homologues of carboxylic acids ($C_5$-$C_{10}$) and 2-alkanones ($C_8$-$C_{13}$). However, although the blends of carboxylic acids and that of 2-alkanones contributed incrementally to the repellency of the waterbuck odor to savannah tsetse flies, some waterbuck constituents (particularly, nonanoic acid and 2-nonanone) showed significant attractive properties. In another study, increasing the ring size of δ-octalactone from six to seven membered ring changed the activity of the resulting molecule (ε-nonalactone) on the savannah tsetse flies from repellency to attraction. In the present study, we first compared the effect of blending ε-nonalactone, nonanoic acid and 2-nonanone in 1:1 binary and 1:1:1 ternary combination on responses of *Glossina pallidipes* and *Glossina morsitans morsitans* tsetse flies in a two-choice wind tunnel. The compounds showed clear synergistic effects in the blends, with the ternary blend demonstrating higher attraction than the binary blends and individual compounds. Our follow up laboratory comparisons of tsetse fly responses to ternary combinations with different relative proportions of the three components showed that the blend in 1:3:2 proportion was most attractive relative to fermented cow urine (FCU) to both tsetse species. In our field experiments at Shimba Hills game reserve in Kenya, where *G. pallidipes* are dominant, the pattern of tsetse catches we obtained with different proportions of the three compounds were similar to those we observed in the laboratory. Interestingly, the three-component blend in 1:3:2 proportion when released at optimized rate of 13.71mg/h was 235% more attractive to *G. pallidipes* than a combination of POCA (3-*n*-Propylphenol, 1-Octen-3-ol, 4-Cresol, and Acetone) and fermented cattle urine (FCU). This constitutes a novel finding with potential for downstream

**Data Availability Statement:** All relevant data are within the manuscript and its Supporting Information files.

**Funding:** POM received funding from the Fogarty International Center of the National Institutes of Health awards R03TW009444, D43TW007391 and U01AI115648. BMW received funding from the National Research Fund Kenya (Government of Kenya) (2016/2017). The funders had no role in study design, data collection and analysis, decision to publish, or preparation of the manuscript.

**Competing interests:** The authors have declared that no competing interests exist.

deployment in bait technologies for more effective control of *G. pallidipes*, *G. m. morsitans*, and perhaps other savannah tsetse fly species, in 'pull' and 'pull-push' tactics.

## Author summary

In our previous studies with tsetse fly-refractory waterbuck body odor, we found that certain subtle structural changes are associated with shifts in activities of some constituents from repellency to attraction. This led us to discovery of three potent tsetse attractants (ε-nonalactone, nonanoic acid and 2-nonanone). In the present study, we explored possible synergistic effects of blending of these compounds in different proportions to *Glossina pallidipes* and *Glossina m. morsitans* in the laboratory, followed by field studies with *G. pallidipes*. A three-component blend comprised of ε -nonalactone, nonanoic acid and 2-nonanone in 1:3:2 proportion gave 235% higher tsetse fly catches in the field compared with that of POCA and FCU. Thus, dispensing this odor blend in tsetse fly traps or insecticide treated targets is expected to suppress the tsetse flies more efficiently. It will also be interesting to see if the blend is similarly attractive to other savannah tsetse fly species.

## Introduction

Artificial visual-olfactory bait technologies have shown significant promise in tsetse fly control operations[1–3]. This is because of their relatively high specificity, low cost, community acceptability, ability to stem tsetse re-invasion from adjacent areas[1–4] and minimal environmental contamination[1]. These technologies are based on long-range (60–120 m) behavioral olfactory responses of tsetse flies to blends of synthetic versions of some natural mammalian host odors and closer range (~10 m) visual attraction that are designed to mimic those from their natural hosts in the field[5]. However, tsetse flies show gradation of preferences for different mammals, with some specific chemical 'fingerprints' playing important roles in locating preferred hosts[6–8], and others in active avoidance of non-hosts[9,10]. Savannah tsetse fly species, including *Glossina morsitans morsitans* and *Glossina pallidipes*, are preferentially attracted to chemical 'signatures' from ungulates and other large mammals, among which buffalo and cattle are most attractive[7,11].

Chemical analyses (with Gas chromatography linked with electroantennographic detector and Gas chromatography linked with mass spectrometer) of these 'signatures' in breath volatiles of buffalo and cattle identified carbon dioxide ($CO_2$), acetone, 2-butanone, 1-octen-3-ol (octenol) as key constituents attractive to *G. m. morsitans and G. pallidipes*[12]. Various blends consisting of these compounds significantly improve performance of traps on savannah tsetse. However, the attraction efficiency has been found to be much less than a fifth of that of preferred hosts (buffalo or cattle)[13]. This suggested involvement of additional kairomones from these hosts. Further studies revealed that buffalo urine was more attractive to *G. pallidipes* than the blend associated with host breath[14], especially when fermented for a couple of days[15], suggesting that the urine harbored other components responsible for the enhanced attractancy of the natural hosts[13]. Field bioassay-guided isolation and characterization of the fermented urine identified a phenolic blend (phenol, 3- & 4-cresols, 3- & 4-ethylphenols and 3- & 4-*n*-propylphenols), and specifically, combination of 4-cresol and 3-*n*-propyl phenol, as key attractants of the flies in the urine[16,17]. Subsequently, a 1:4:8 blend of the 3-*n*-propylphenol, octenol, and *p*-cresol together with separately released acetone (collectively referred to as POCA)

was evaluated and established as more attractive to *G. m. morsitans* or *G. pallidipes*[18,19]. However, POCA was found to have about 25% attraction relative to odors of natural hosts (cattle or buffalo)[13]. This underlined the need for further search for more potent odor attractant blends with either incremental effects on POCA or better intrinsic attraction.

The present study was built on results from two sets of studies with constituents of tsetse fly-refractory waterbuck. First, wind tunnel and field studies with tsetse flies (*G. pallidipes* and *G. m. morsitans)* showed that although the blend of carboxylic acids and that of 2-alkanones contributed incrementally to the repellency of the waterbuck odor to, there was redundancy within each of the two groups, with some constituents (particularly, octanoic acid, nonanoic acid and 2-nonanone) demonstrating significant attractive properties[20]. Second, structure-activity studies with different closely related analogues of δ-octalactone showed that increasing the ring size from six to seven membered ring changed the activity of the resulting molecule (ε-nonalactone)[21] on the savannah tsetse flies from repellency to attraction[21]. In this study, we explored the effects of blending three attractive compounds, i.e. ε-nonalactone, nonanoic acid (which was found to be more attractive than octanoic acid) and 2-nonanone in different combinations and proportions to *G. pallidipes* and *G. m. morsitans* compared to that of the phenolic blend of fermented cow urine in a 2-choice wind tunnel. We then compared the performance of selected blends with a combination of POCA and fermented cattle at Shimba Hills game reserve in Kenya, where *G. pallidipes* are dominant.

## Materials and methods

### Test chemicals

We sourced 2-nonanone, nonanoic acid and dichloromethane (98–99%) from Sigma-Aldrich, Taufkirchen, Germany. We synthesized ε-nonalactone in the laboratory following the method of Gikonyo et al. [9] as modified by Wachira et al. [21]. The ε-nonalactone synthesized consisted of a racemic mixture of equal quantities (50: 50) of (+) ε-nonalactone and (-) ε-nonalactone enantiomers as described by Wachira et al. [21].

We confirmed the structure of the resultant ε-nonalactone using spectra of the molecule generated from Infra-Red (IR) Spectrometer (Shimadzu, Kyoto, Japan), High Resolution Mass Spectrometer (HR-MS) (Jiangsu Skyray Instrument Co., Ltd., Kunshan, China), Proton ($^1$H- NMR) and Carbon-13 ($^{13}$C-NMR) Nuclear Magnetic Resonance Spectrometer (Agilent Technologies, Inc., California, USA), as outlined by Skoog et al. [22].

### Laboratory bioassays for behavioral responses of *G. m. morsitans* or *G. pallidipes* to compounds and blends

We conducted laboratory studies with both sexes of adult *G. m. morsitans* or *G. pallidipes* obtained from colonies maintained at the insectary at Biotechnology Research Institute of Kenya Agricultural and Livestock Research Organization (BioRI-KALRO), Muguga, Kenya. The two tsetse colonies were established from seed puparia material previously received from International Atomic Energy Agency (IAEA) laboratories, Seibersdorf, Austria colonies, which originated from *G. m. morsitans* wild pupae collected from Zimbabwe in 1983 or *G. pallidipes* wild pupae collected from Lugala, Uganda in 1975[21]. The flies were reared at 25 ± 2˚C, 75 ± 2% relative humidity and 12:12 h light-day photoperiod in the insectary, and were fed with defibrinated bovine blood through artificial silicon-membrane based feeding devices[23] three times a week.

For assessment of behavioral responses of G. *m. morsitans* or *G. pallidipes* to synthetic odors, we prepared the following blends: A (2-nonanone + nonanoic acid), B (2-nonanone +

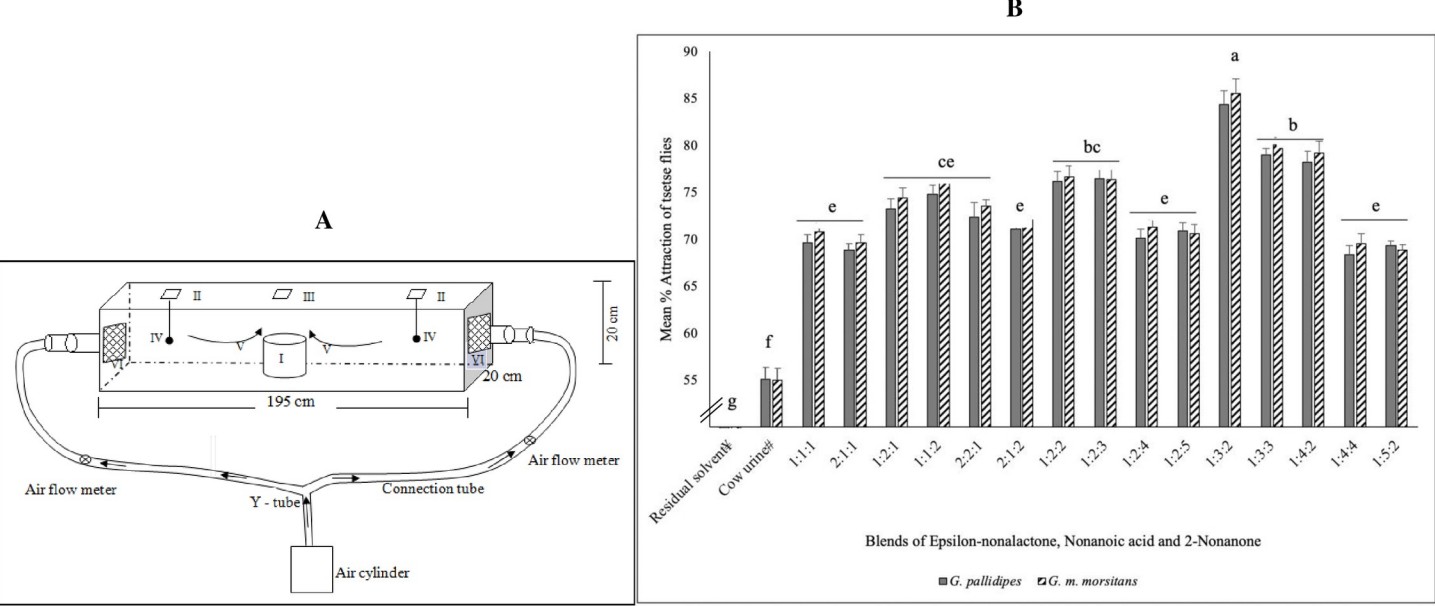

**Fig 1. A:** Schematic diagram of the wind tunnel employed for behavioral bioassay Insect release cage (**I**), windows for introducing test material dispensers (**II**), window for introducing insect release cage (**III**), test material dispenser (**IV**), air flow direction (**V**) and PVC gauze for keeping flies within the tunnel (**VI**). **B:** Wind-tunnel responses of *G. pallidipes* and *G. m. morsitans* to attractive blends in varying proportions. Means followed by the same letter are not significantly different ($p>0.05$, SNK post-hoc analyses).

ε-nonalactone), C (ε-nonalactone + nonanoic acid) and D (ε-nonalactone + nonanoic acid + 2-nonanone). Each blend consisted of equal proportion of each of the individual constituents. We monitored responses of the flies to individual chemicals and each of the four blends in a plexi glass cuboidal wind tunnel (195 cm × 20 cm × 20 cm) (Fig 1A) following protocols previously described (Gikonyo et al. [9]; Wachira et al. [21]).

Briefly, we placed 30 three-day old teneral *G. pallidipes* or *G. m. morsitans* in a release cage (indicated as **I** in Fig 1A) and allowed them to acclimatize for ten minutes. We carried out assessment of responses of the flies to each compound (2-nonanone, ε-nonalactone, octanoic acid or nonanoic acid) and blend (A, B, C or D) at concentrations of 0.05, 0.25 and 0.5 mg of each compound or blend in 1 ml dichloromethane solvent. We separately released the prepared compounds/blends from one side of the olfactometer and dichloromethane solvent from the other side both at 12.6 l/min[21]. During three-minute observations we monitored, 1) number of flies departing from midsection, 2) initial direction of upwind flights, and 3) final landing and resting positions distance (in cm) from the midsection release point in either direction (control or treated arms) in the wind tunnel). Each arm of the wind tunnel was graduated in cm from the midsection to facilitate measurement of flight distances. We assessed responses to each dose of each compound or blend in triplicates. At the end of each cycle of observations, we removed the flies from the wind tunnel using insect mechanical aspirator. To minimize cross-contamination between experiment cycles, we cleaned the wind tunnel, metallic racks and release cages first with distilled water and then with 70% absolute ethanol in distilled water, and then passed air in the tunnel at high speed of about 20l/min for 15 minutes. We confirmed absence of residual effects of previously tested odors by running blank tests (consisting of the solvent in both odor dispensers) and assessing responses of the flies. In addition, we tested each compound or blends on different days, and alternated control and odor arms of the tunnel in successive replicates to minimize directional biases. We tested each

compound or blend in three replicates in the mornings (0800–1200 hrs) or evenings (1500–1700 hrs). These periods are consistent with periods of peak biological activities of *G. pallidipes* or *G. m. morsitans*[24,25]. For each compound and blend, we established relative choice preference index, defined as $((\textcircled{T}-\textcircled{C})/(\textcircled{T}+\textcircled{C}))^{*}100$, where $\textcircled{T}$ and $\textcircled{C}$ represented the number of flies in the treated and control arms, respectively[10]. We evaluated each index by comparing average distances of upwind flight by activated flies in treated and control arms using Chi Square ($\chi^2$) non-parametric test. We then rank-transformed the proportional responses of flies to various odors, analyzed the transformed data using analysis of variance (ANOVA) and separated the means using LSD post hoc analysis.

### Field bioassay validations of wind tunnel for behavioral responses of *G. pallidipes* to odors

We limited our field validation to *G. pallidipes*, excluding *G. m. morsitans* that was not present in tsetse fly habitats in our region (East Africa). The field *G. m. morsitans* was therefore not available to us for validation at that point.

Our wind tunnel results provided us with general information on choice preferences of the flies to the selected odors that could have been affected by 1) number of generations of the flies that had been colonized which could affect its phenotypic behavior, and 2) rate of air flow in the wind tunnel and its possible effect on the responses of the flies, given that tsetse flies fly at 5 m/s, with a minimum of 2.5 and a maximum of 7 m/s in the field[24,25]. We thus evaluated all the compounds and blends tested in the wind tunnel in the field, except octanoic acid, which showed lower attraction compared to nonanoic acid. We conducted the responses validation experiments against adult *G. pallidipes* at Shimba Hill National Reserve (-4˚ 15' 26" S, 39˚ 23' 16" E; altitude 403 m) in Kwale County, Kenya where *G. pallidipes* are naturally abundant. The *G. m. morsitans* were absent in this study site; however, given their similar laboratory and field responses to odors in previous experiments[21], they can be assumed to show comparable field responses. The reserve occupies 300 Km$^2$ and is inhabited by mammals that include hartebeest, sable antelopes, buffalos, elephants, bushbucks, bush pigs, warthogs, leopards, giraffes, monkeys and duikers, among which buffalos and bushbucks are most preferred by *G. pallidipes*[10]. The vegetation consists mostly of coastal rainforest and semi-evergreen woodland and grassland. Shimba Hill National Reserve has an annual rainfall of 855–1682 mm and temperature of 24.2˚C. Most of the rainfall is experienced from April to June (long rain season) and October to November (short rain season). Highest daily temperatures ($\approx$31˚C) are experienced in March and November and lowest temperatures ($\approx$27˚C) in July of each year. We conducted these experiments in December, after the short rains, when the *G. pallidipes* populations were most abundant.

We assessed the responses of *G. pallidipes* to the odors/blends using Latin Square field experimental design as previously described in Wachira et al. [21]. This experimental design allowed us to discount the confounding effects of site and days of experiments from those of our treatments in subsequent analysis and interpretation of our results. Briefly, we deployed NG2G traps[15] on sites that were about 300 meters apart. Since *G. pallidipes* can detect and track odors from at most 100 meters[15], we considered our sites sufficiently spaced to minimize interactions between the treatments. The sites had similar level of *G. pallidipes* densities of 200–240 flies/trap/day. We separately baited each trap on site with fermented cow urine (FCU) ($\approx$ 1000mg/h), acetone ($\approx$500mg/h) and a single odor or blend (19.33–22.00mg/h), or no odors (control). Hence, we deployed traps and baited them separately with nonanoic acid, ε-nonalactone, 2-nonanone, blends A, B, C or D. We dispensed the odors (4 mls per pack) using sealed thin-walled polythene lay flat tubing of 150 microns thickness (folded into

tetrahedrons forming a sachet with a surface area of 50 cm$^2$). We established release rates of the compounds/blends from lost masses of the sachets with respective compound or blend after every 24 hrs. We also evaluated effect of release rate on the catches of tsetse by using lay flat tubing of different thickness and varying number of sachets deployed per traps. In all cases, we placed the sachets on split metallic rods and pegged on the ground (to minimize site contamination) $\approx$30 cm downwind side of the traps. We also baited one of the sites with FCU and acetone and another one with no odor as controls. In a series of randomized Latin Squares Design experiments in three independent blocks about two kilometers apart that constituted our replicates, we deployed the baits at about 9:00 am of each day rotated the baits at 24 hrs intervals to accommodate both peaks (morning and evening) of *G. pallidipes* activities[24,25]. We collected tsetse and all other insects in each trap at the end of each interval, sorted and identified them according to their tsetse species and non-target insect genera using taxonomic keys[26,27]. We further evaluated performance of odor/blend with the highest *G. pallidipes* attraction relative to POCA[18], the standard tsetse attractant combination used in routine tsetse fly control.

In our data analyses, we normalized the distributions and homogenized the variances in our catches of *G. pallidipes* by log(n+1) transformation of the catch numbers. We then analyzed the transformed data using one-way analysis of variance (ANOVA) with day, site and treatments as factors. We separated the means using LSD post hoc analyses. We then back-transformed (antilog) the data for reporting. We further transformed the catches of each into indices, relative to the trap baited with FCU and acetone. We analyzed all our laboratory and field data using SPSS Version 22 (SPSS, Inc., Chicago, IL, U.S.A.) with significance level set at 5%.

## Results

### Laboratory wind tunnel responses of *G. m. morsitans* and *G. pallidipes* to tsetse attractant compounds and blends

We confirmed structure of ε-nonalactone (**1**) synthesized using (HR-MS), C-13 Nuclear Magnetic Resonance ($^{13}$C-NMR), proton Nuclear Magnetic Resonance ($^1$H-NMR), and Infrared (IR) spectroscopic techniques as evidenced by the spectra (S1 Fig).

We have summarized Wind tunnel laboratory responses of *G. m. morsitans* and *G. pallidipes* in Table 1. We observed similar response patterns to the different compounds/blends at the three doses, so we have provided only those of 0.25 mg in 1 ml dichloromethane and doses required to achieve the desired effect in 75% of the flies (ED75%) for *G. pallidipes* or *G. m. morsitans* (Table 1). In both species, most (>90%) flies were activated to leave the mid-section when introduced into the wind tunnel. Overall, flight distance and final resting choice response pattern were similar for the two tsetse fly species (Table 1), although *G. m. morsitans* were generally more responsive than *G. pallidipes* to the test compounds or blends (Table 1). Flies flew significantly ($p<0.05$) longer distances in the arms with odor compounds or blends relative to the arms with no-odor control, except in the presence of δ-octalactone a known tsetse repellent used as a negative control, which led to negative attractancy (repellency) both *G. pallidipes* (-41.60%) and *G. m. morsitans* (-46.07%) (Table 1). Blend C or D in the treated arms exhibited significantly higher *G. pallidipes* preference than odors from fermented cow urine treatments (Table 1). Similarly, *G. m. morsitans* exhibited significantly ($p<0.05$) greater final resting choice preferences for blends B, C or D than odors from fermented cow urine control (Table 1). Resting choice preferences in both species for the other compounds and blends were similar or less than the observed preferences for fermented cow urine (Table 1).

The attraction of both species to blends A, B, C or D was significantly higher than each of the individual constituent compounds, except with *G. m. morsitans* to blend A, where

**Table 1. Laboratory behavioral responses of *G. pallidipes* and *G. m. morsitans* in a wind tunnel to selected compounds/blends related to waterbuck.**

| Dose (mg per 1ml solvent) | Test material | *G. pallidipes* | | | *G. m. morsitans* | | |
|---|---|---|---|---|---|---|---|
| | | Average distance of upwind flight (cm ± SE) | | Final resting choice | ED75% (mg) | Average distance of upwind flight (cm ± SE) | | Final resting choice | ED75% (mg) |
| | | C | T | $((T-C)/(T+C))^* \times 100 \pm SE$ | | C | T | $((T-C)/(T+C))^* \times 100 \pm SE$ | |
| 0 | Residual solvent¥ | 58.52 ± 6.31 | 54.34 ± 4.47 | 0.00 ± 0.00h | - | 61.31 ± 4.11 | 48.49 ± 3.71 | 0.00 ± 0.00f | - |
| | Cow urine# | 33.39 ± 6.08 | 67.80 ± 5.36* | 59.43 ± 0.71cd | - | 42.09 ± 3.42 | 73.02 ± 6.25 | 55.61 ± 1.24de | - |
| 0.25 | Delta-octalactone (repellent; negative control) | 56.34 ± 2.70 | 28.33 ± 3.87** | - 41.60 ± 1.50i | 0.57 | 59.25 ± 9.58 | 25.26 ± 3.10*** | - 46.07 ± 1.00g | 0.53 |
| | 2-Nonanone | 30.63 ± 5.76 | 44.90 ± 4.56 | 45.03 ± 2.23f | 0.55 | 28.14 ± 2.94 | 54.76 ± 5.57 | 50.37 ± 5.19e | 0.45 |
| | Nonanoic acid | 22.57 ± 7.26 | 57.00 ± 4.86** | 51.85 ± 2.06e | 0.46 | 20.00 ± 4.80 | 63.57 ± 5.17 | 58.52 ± 1.48d | 0.39 |
| | Octanoic acid | 29.80 ± 3.95 | 40.65 ± 3.78 | 33.33 ± 0.67g | 0.62 | 31.44 ± 4.82 | 49.05 ± 4.63 | 37.78 ± 2.22g | 0.58 |
| | Epsilon-nonalactone | 37.34 ± 4.71 | 58.63 ± 3.56 | 52.43 ± 1.96e | 0.46 | 34.99 ± 5.54 | 60.37 ± 6.98 | 56.31 ± 1.90d | 0.43 |
| | **Blend A** (2-Nonanone + Nonanoic acid) | 33.17 ± 4.00 | 58.65 ± 1.43* | 56.36 ± 0.07de | 0.44 | 30.39 ± 1.94 | 61.18 ± 3.22** | 58.04 ± 0.01d | 0.41 |
| | **Blend B** (2-Nonanone + Epsilon-nonalactone) | 34.04 ± 2.85 | 59.79 ± 2.67* | 61.24 ± 1.41bc | 0.39 | 28.99 ± 1.87 | 60.79 ± 4.09** | 63.09 ± 1.01c | 0.4 |
| | **Blend C** (Epsilon-nonalactone + Nonanoic acid) | 26.94 ± 1.68 | 65.64 ± 3.36*** | 67.84 ± 1.65b | 0.32 | 22.81 ± 2.70 | 66.84 ± 2.00*** | 68.90 ± 1.14b | 0.32 |
| | **Blend D** (Epsilon-nonalactone + Nonanoic acid + 2-Nonanone) | 17.32 ± 2.01 | 67.76 ± 2.71*** | 74.58 ± 1.11a | 0.25 | 15.33 ± 2.39 | 71.12 ± 2.74*** | 75.12 ± 1.11a | 0.27 |

Number of tsetse flies used in each test (N = 30 × 3); C = control arm; T = treated arm; © = Number of flies in C; T = Number of flies in T

#3-days fermented cow urine

¥dichloromethane solvent used to dissolve the test compounds, Each pair of average distance of upwind flight in C and T were compared by χ2 (**p < 0.01;

***p < 0.001); means followed by the same letter in final resting choice are not significantly different ($p > 0.05$, SNK post-hoc test); ED75% effective dose that can attract/repel three-quarters of tsetse population.

2-nonanone seemed to have no significant effect when blended with nonanoic acid (Table 1). $ED_{75}$ for the three-component blend D was significantly lower than those of other compounds and blends in both species (0.25 mg and 0.27 mg for *G. pallidipes* and *G. m. morsitans*, respectively) (Table 1).

Tsetse fly responses to blend D (ε-nonalactone + nonanoic acid + 2-nonanone) in varying proportions of the constituent compounds led to a more effective composite consisting of one part of epsilon-nonalactone, three parts of nonanoic acid, and two parts of 2-nonanone (1:3:2 respectively) (Fig 1B). An increase of the proportion of any of the constituents relative to the 1:3:2 blend, led to significant ($p < 0.05$) decreases in tsetse response toward the resultant blend, as observed with 1:3:3, 1:4:2, 1:4:4 and 1:5:2 blends (Fig 1B).

## Field responses of *G. pallidipes* to baited NG2G traps in the presence of selected attractive compounds/blends

The *G. pallidipes*, *G. austeni* and *G. brevipalpis* are sympatric at Shimba Hills game reserve, with *G. pallidipes* most predominant, constituting 95.42% of the overall numbers of flies that responded to treatments. The *G. austeni* and *G. brevipalpis* constituted 2.83 and 0.49% respectively. Consequently, the responses of *G. austeni* and *G. brevipalpis* were about 12 and 4 flies per trap per day, which we considered insufficient for us to draw relevant statistical inference

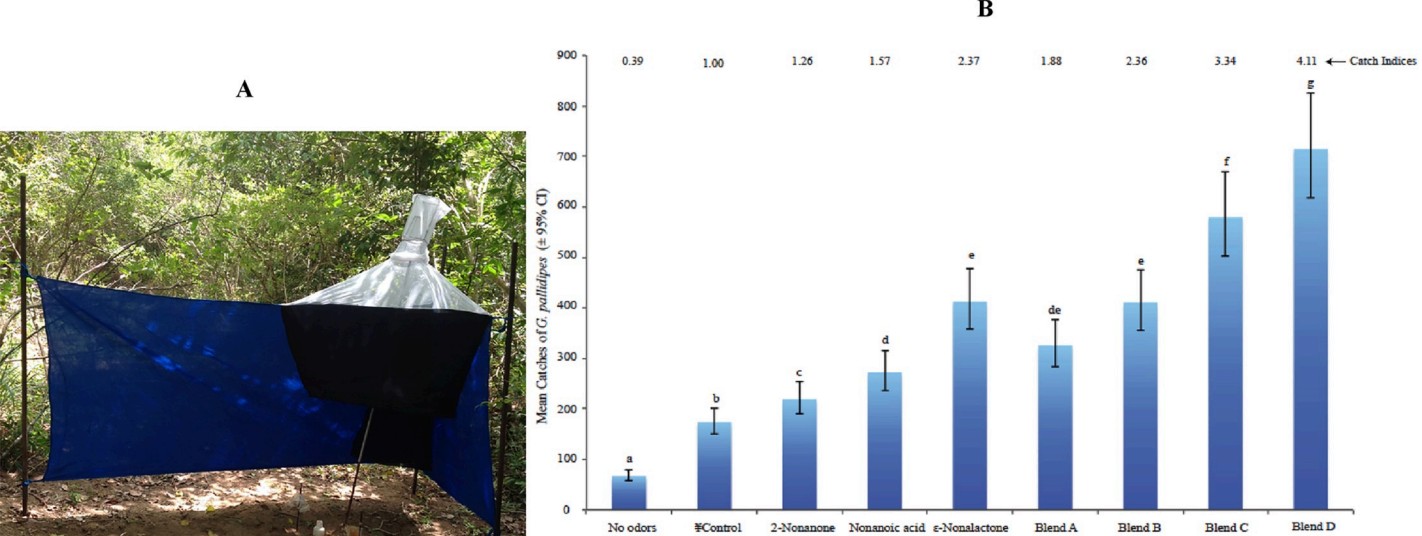

**Fig 2. A:** A photograph of NG2G trap for evaluating relative attraction of tsetse flies to chemical odors. **B:** Relative mean field catches of *G. pallidipes* to attractant baited NG 2 G traps. ¥Trap baited with fermented cow urine and POCA; means followed by the same letter are not significantly different (*p*>0.05, LSD post-hoc analyses); Catch index = total mean catches expressed as proportion of that of control trap.

regarding responses of the respective species to our treatment. We thus excluded these two species from further analyses.

Analysis of catches of baited NG2G traps in the field (Fig 2A) showed that *G. pallidipes* were attracted to the odor blends with a pattern that revealed clear odor specificity (Fig 2B). In general, presence of odors significantly enhanced the trap catches in similar patterns to those observed in the wind-tunnel experiments. All traps with test odors or blends had significantly more tsetse catches than traps with either standard attractant blend control (POCA), or no odor control traps (Fig 2B). The test odors had better performance in blends than in individual component formulations, as observed in wind tunnel assays (Fig 2B).

Traps baited with ε-nonalactone or blend D had the most catches among traps baited with individual odors or blends respectively (Fig 2B). The catches in traps baited with ε-nonalactone were similar to those baited with either of the 2-nonanone-containing blend A or B, but was significantly (*p*<0.05) lower than the nonanoic acid containing blends C or D (Fig 2B). Thus, ε-nonalactone and 2-nonanone showed the highest and lowest incremental impact, respectively, on catches of the baited traps in the blends. The odor compounds/blends were dispensed in polythene lay flat tubing of 150 microns thickness, and the release rate of individual compounds and blends (18.43–22.01 mg/h) in the field were similar (*p*>0.05) (Table 2).

With sachet thickness of 150 microns, the three-component blend of epsilon-nonalactone, nonanoic acid and 2-nonanone (in 1:3:2 respective proportions) gave higher *G. pallidipes* catches with 235% increase as compared to those of the baited control and the 1:1:1 trap (Table 3).

## Discussion

The results of our present study confirm that very subtle structural changes in some of the constituents of waterbuck body odour (lactone, 2-alkanone and carboxylic acid) are associated with shifts in their activities from repellence to attraction to the tsetse flies. In addition, in a wind tunnel study, two or three component blends of ε-nonalactone, nonanoic acid and 2-nonanone, show varying levels of synergistic effects between the compounds, with the three-

**Table 2. Field release rate of odor compounds/blends from 150 microns sachet.**

| Test Compound | Mean (mg/h) | 95% Confidence Interval |
|---|---|---|
| Epsilon-nonalactone | 22.01 | 21.39–22.61 |
| Nonanoic acid | 19.93 | 17.58–22.28 |
| 2-Nonanone | 17.67 | 12.32–23.02 |
| Blend A | 19.01 | 17.38–20.64 |
| Blend B | 18.43 | 16.16–20.70 |
| Blend C | 18.65 | 15.24–22.06 |
| Blend D | 18.86 | 15.88–21.84 |
| POC | 18.12 | 15.42–22.21 |
| *p*—value | > 0.05 | |

Blends A (2-nonanone + nonanoic acid), B (2-nonanone + epsilon-nonalactone), C (epsilon-nonalactone + nonanoic acid) and D (epsilon-nonalactone + nonanoic acid + 2-nonanone), with each blend consisting of equal proportions of its constituents; POC = a three-component blend comprising of **P** = 3-n-propylphenol, **O** = 1-octen-3-ol and **C** = 4-methylphenol (p-cresol); Acetone released from a bottle at 489.3 mg/h.

component blend of the lactone, ketone and carboxylic acid in 1:3:2 relative proportion being most attractive to both *G. pallidipes* and *G. m. morsitans*. In the field study, where *G. pallidipes* is dominant, the three compounds and blends showed similar pattern of responses to the blends, and the 1:3:2 blend released at an optimized rate (18.43–22.01 mg/h) was 2.35 times more attractive than POCA (P = 3-n-propylphenol; O = 1-octen-3-ol; C = 4-methylphenol; A = acetone) combination. Thus, our three-component blend appears to constitute a significantly more potent attractant with potential for downstream deployment as a more effective bait for mass trapping of savannah tsetse flies in the field.

Three follow up studies are currently planned. First, the new three-component blend needs to be compared with POCA in a semi-field study with individual cattle to confirm its enhanced attractiveness. Second, it will be interesting to see if individual POCA constituents and different blends have incremental attractive effects on the three-component blend, and if a new combination can be derived with level of attraction comparable to that of individual cattle. Finally, field comparison of the *G. pallidipes* and *G. m. morsitans* might confirm the laboratory responses or reveal subtle differences, especially where both flies are sympatric, which was not possible in the current study.

The results of this study lay down useful groundwork in the large-scale development of more effective tsetse baits to be used in 'pull' and 'pull-push' control tactics.

**Table 3. Mean catches of *G. pallidipes* to NG2 G traps in the presence of blends of epsilon-nonalactone, nonanoic acid and 2-nonanone in varying proportions delivered from 150micron sachets.**

| Blend in various proportions | Mean catches ± SE | 95% Confidence Interval | #Catch index | % Increase |
|---|---|---|---|---|
| 1:1:1 | 81.09[d] | 66.88–95.30 | 1.28 | 28 |
| 1:3:2 | 212.23[a] | 186.52–247.94 | 3.35 | 235 |
| 1:4:2 | 160.29[b] | 150.06–160.52 | 2.53 | 153 |
| 1:2:2 | 118.85[c] | 96.86–140.84 | 1.88 | 88 |
| ¥Control | 63.28[e] | 43.17–83.39 | 1 | |

[¥]Trap baited with POCA; means followed by the same letter are not significantly different (*p*>0.05, LSD post-hoc analyses)

[#]Total mean catch expressed as proportion of that of the control trap.

In addition, *G. pallidipes* catches appeared to increase with increased rate of release of the 1:3:2 blend around the NG2 G traps (Table 4).

**Table 4. Mean catches of *G. pallidipes* to optimized blend of epsilon-nonalactone, nonanoic acid and 2-nonanone (1:3:2) dispensed at various release rates.**

| Sachet thickness (microns) | Number of sachets per trap | Release rate (mg/h) | Mean *G. pallidipes* catches | #Catch index |
|---|---|---|---|---|
| 150 | 1 | 4.57[f] | 36.7[e] | 1.4 |
| 100 | 1 | 6.34[e] | 90.1[c] | 3.4 |
| 150 | 2 | 9.12[d] | 75.4[d] | 2.8 |
| 65 | 1 | 10.82[c] | 136.8[b] | 5.1 |
| 150 | 3 | 13.71[b] | 156.6[a] | 5.8 |
| Control trap (POCA) | 1 | 18.12[a] (POC) and Acetone (489.3) | 26.8[f] | |
| No odor trap | - | - | 17.3[g] | 0.6 |

Means followed by different letters are significantly ($p<0.05$; ANOVA and LSD post-hoc analyses) different

#Total mean catch expressed as proportion of that of the control trap.

## Rights and permissions

Field study was carried out in Shimba Hills National Reserve with permission from Kenya Wildlife Service (reference KWS/BRM/5001 dated 4[th] April, 2018) under the condition that a research fee was paid. Additionally, we work closely and share reports and findings/publications with their senior scientist in Biodiversity Research and Monitoring. The permit is given to the project under the title 'Control of Tsetse Fly Transmitted Diseases in Kenya'.

## Supporting information

**S1 Fig. Spectral confirmation of synthesized epsilon-nonalactone.** A: High Resolution–Mass Spectrometer spectrum. B: Exact molecular masses leading to identification. C: Carbon-13 Nuclear Magnetic Resonance spectrum. D: Proton (H) Nuclear Magnetic Resonance spectrum. E: Infra-red spectrum.
(TIF)

## Acknowledgments

We thank Dr. Rosemary Bateta (HOD Entomology of BioRI-KALRO) for providing facilities and technical support in laboratory and field studies.

## Author Contributions

**Conceptualization:** Benson M. Wachira, Joy M. Kabaka, Paul O. Mireji, Sylvance O. Okoth, Margaret M. Nganga, Grace A. Murilla, Ahmed Hassanali.

**Data curation:** Benson M. Wachira, Joy M. Kabaka, Paul O. Mireji, Sylvance O. Okoth, Margaret M. Nganga, Patrick Obore.

**Formal analysis:** Benson M. Wachira, Paul O. Mireji, Grace A. Murilla, Ahmed Hassanali.

**Funding acquisition:** Paul O. Mireji, Ahmed Hassanali.

**Investigation:** Paul O. Mireji, Patrick Obore.

**Methodology:** Benson M. Wachira, Joy M. Kabaka, Sylvance O. Okoth, Margaret M. Nganga, Robert Changasi, Bernard Ochieng'.

**Project administration:** Paul O. Mireji, Sylvance O. Okoth, Grace A. Murilla.

**Resources:** Margaret M. Nganga, Robert Changasi, Patrick Obore, Bernard Ochieng', Grace A. Murilla.

**Software:** Benson M. Wachira.

**Supervision:** Paul O. Mireji, Margaret M. Nganga, Grace A. Murilla, Ahmed Hassanali.

**Visualization:** Robert Changasi, Bernard Ochieng'.

**Writing – original draft:** Benson M. Wachira, Joy M. Kabaka.

**Writing – review & editing:** Benson M. Wachira, Paul O. Mireji, Margaret M. Nganga, Grace A. Murilla, Ahmed Hassanali.

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
