## [Decision Letter · Decision Letter 0]

8 Jun 2020

Dear Dr. Wachira,

Thank you very much for submitting your manuscript "Odor blend with enhanced attraction of selected savannah tsetse fly vectors of African trypanosomiasis" for consideration at PLOS Neglected Tropical Diseases. As with all papers reviewed by the journal, your manuscript was reviewed by members of the editorial board and by several independent reviewers. In light of the reviews (below this email), we would like to invite the resubmission of a significantly-revised version that takes into account the reviewers' comments. 

We cannot make any decision about publication until we have seen the revised manuscript and your response to the reviewers' comments. Your revised manuscript is also likely to be sent to reviewers for further evaluation.

Sincerely,

Rhoel Ramos Dinglasan

Associate Editor

Alvaro Acosta-Serrano

Deputy Editor

Reviewer's Responses to Questions

**Key Review Criteria Required for Acceptance?**

**Methods**

-Are the objectives of the study clearly articulated with a clear testable hypothesis stated?

-Is the study design appropriate to address the stated objectives?

-Is the population clearly described and appropriate for the hypothesis being tested?

-Is the sample size sufficient to ensure adequate power to address the hypothesis being tested?

-Were correct statistical analysis used to support conclusions?

-Are there concerns about ethical or regulatory requirements being met?

Reviewer #1: Comments on the methods is provided in the detail review report

Reviewer #2: See attachment

**Results**

-Does the analysis presented match the analysis plan?

-Are the results clearly and completely presented?

-Are the figures (Tables, Images) of sufficient quality for clarity?

Reviewer #1: Results are well written with few comments highlighted in the attached detailed report

Reviewer #2: see attachment

**Conclusions**

-Are the conclusions supported by the data presented?

-Are the limitations of analysis clearly described?

-Do the authors discuss how these data can be helpful to advance our understanding of the topic under study?

-Is public health relevance addressed?

Reviewer #1: The conclusions drawn is representation of the results finding and good enough to support the finding

Reviewer #2: Data reanalysis is necessary to support the conclusions from the laboratory portion of the study (see attachment)

**Editorial and Data Presentation Modifications?**

Reviewer #1: Data is presented well and few things pointed out in the detail report for the attention of the authors

Reviewer #2: The supplemental information needs to be presented in a clearly viewable fashion (see attached).

**Summary and General Comments**

Reviewer #1: This research evaluates the synergistic effects of known molecules from body odor of waterbuck consisting of delta-octalactone, geranyl acetone, phenols, C5-C10 carboxylic acids and C8-C13 2-alkanones that could be use for tsetse flies control by national programmes, communities, military and wildlife centres. I recommend this paper to be published to support future data in this line of identification of molecules for tsetse control.

Reviewer #2: The paper presents an interesting new odor blend that could be useful for Tsetse fly control. The data needs further analysis to support all of the conclusions made in the paper.

PLOS authors have the option to publish the peer review history of their article (what does this mean?). If published, this will include your full peer review and any attached files.

Reviewer #1: No

Reviewer #2: No
---

## [Decision Letter · Decision Letter 1]

29 Mar 2021

Dear Dr. Wachira,

Thank you very much for submitting your manuscript "Characterization of a composite with enhanced attraction to savannah tsetse flies from constituents or analogues of tsetse refractory waterbuck (Kobus defassa) body odor" for consideration at PLOS Neglected Tropical Diseases. As with all papers reviewed by the journal, your manuscript was reviewed by members of the editorial board and by several independent reviewers. The reviewers appreciated the attention to an important topic. Based on the reviews, we are likely to accept this manuscript for publication, providing that you modify the manuscript according to the review recommendations. 

Sincerely,

Rhoel Ramos Dinglasan

Associate Editor

Alvaro Acosta-Serrano

Deputy Editor

Reviewer's Responses to Questions

**Key Review Criteria Required for Acceptance?**

**Methods**

-Are the objectives of the study clearly articulated with a clear testable hypothesis stated?

-Is the study design appropriate to address the stated objectives?

-Is the population clearly described and appropriate for the hypothesis being tested?

-Is the sample size sufficient to ensure adequate power to address the hypothesis being tested?

-Were correct statistical analysis used to support conclusions?

-Are there concerns about ethical or regulatory requirements being met?

Reviewer #1: The field and laboratory bioassays are correctly used to test the hypohtesis. The study used Wind tunnel and Latin Square design in evaluate the new blends. These are acceptable methods in these kind of studies. Appropriate statistics were used and presentation of the data is correct.

Reviewer #2: Methods are acceptable

**Results**

-Does the analysis presented match the analysis plan?

-Are the results clearly and completely presented?

-Are the figures (Tables, Images) of sufficient quality for clarity?

Reviewer #1: The results are presented in the logical way and easy to follow.

Reviewer #2: The conclusions are supported by the results. I have comments below to improve the clarity of the data presentation.

**Conclusions**

-Are the conclusions supported by the data presented?

-Are the limitations of analysis clearly described?

-Do the authors discuss how these data can be helpful to advance our understanding of the topic under study?

-Is public health relevance addressed?

Reviewer #1: The conclusion presented is good for data presented more so for G. pallidipes

Reviewer #2: Yes to all.

**Editorial and Data Presentation Modifications?**

Reviewer #1: The paper should be accepted with Minor Revision

Reviewer #2: See summary and general comments.

**Summary and General Comments**

Reviewer #1: The work explore development and evaluation of the alternative attractants for savannah tsetse flies which will go a long way in enhancing traditional trapping technologies for these groups of the tsetse flies. The data presented if good for G. pallidipes. I recommend field bioassay are done on the G. m. morsitans to support the findings from this study. 

Abstract

Line 24 to 25, it should be stated that “The carboxylic acids and alkanones” are also derived from waterbuck 

Line 25, A structural analogue of the δ-octalactone (ε-nonalactone) is also attractive to the “savanna” species of tsetse flies.

Line 32, It’s stated G. pallidipes are naturally abundant in Shimba Hills game reserve. It is clear the result presented support attraction of G. pallidipes to the new blends. The question how did the authors model this to reflect 

G. m. morsitans since these two tsetse types are different?

Introduction

Line 67 to 68, the authors should quote the references source of bait technologies.

Line 97-to 101, needs more specific information to waterbuck odor as shown in POCA justification.

Materials and methods

Line 110, racemic mixture of ε-nonalactone, it might be better to state the proportion of the racemic mixture without referring it to Wachira et al., since this is serious experiment

Laboratory bioassays

Tsetse flies used to evaluate the responses of G. m. morsitans where from the insectary

Field bioassay

Line 205, Our sites had similar G. pallidipes densities, would be possibly to state these densities??

Line 210, they author could also state the release rates

Results

Lab bioassays

Line 236 to 241, racemic mixture of ε-nonalactone ratios are not shown

Line 250 Results for flight distance measurement, it is not very clear in the materials and methods how the flight distance were recorded.

Line 269 to 273, it is not clear what residual solvent was used to get the results shown in Figure 2?

Field bioassays

Line 278, G. pallidipes, G. austeni and G. brevipalpis were attracted. When one checks on the results, there are no results for G. austeni and G. brevipalpis, What do they authors say about this???

Discussion

Line 308 to 309, the authors could expand on the similar receptors and possible genes, if there is some literature in this line. Similar to line 320 to 321

Line 311, subtle structural changes, possibly the authors could look at one thing at a time of increase assessment of the aliphatic and also separately decrease with supporting data. Do authors have these data in the results?

Line 326, The authors have to remember that there was no data for G. austeni and G. brevipalpis populations to support this statement.

Reviewer #2: The authors’ presentation of the discovery of an attractive components of an otherwise repellent odor blend from the Waterbuck is compelling and worth publishing. The combination of laboratory and field work strengthens the conclusions of the paper and could be applied to Tsetse fly control. I do have some suggestions for revision to clarify the manuscript and make it more accessible to a wider readership. The authors have adequately addressed the comments by previous reviewers.

Key points to revise:

The abstract could be more concise and more clearly state the importance of the study. The author summary is clearer than the abstract and highlights the finding better for the reader. The abstract should be revised and the author summary should inspire the revised text.

Line 106: The final sentence is unnecessary, please remove it “Herein we report our findings”

Line 343-344: The sentence “The results of these studies will be reported elsewhere” should be removed. It is unnecessary and does not add anything to the article. Better to add a concluding sentence discussing the relevance of the work towards the development of better attractants for Tsetse flies.

Figures:

Figure 1 & 2 can be combined so it is clear that figure 2 data arises from the assay depicted in figure 1.

Figure 2: Y axis should be set from 0 to 100%. The baseline attraction in the assay to the residual solvent is obscured. Using a dot scatter plot instead of a box plots would more easily allow the reader to assess the N used in each treatment. Color could be used to indicate species. The results are significant and important, but full transparency is necessary for the reader to properly interpret the data.

Figure 3: The resolution of the figure 3 is poor, please use a higher resolution file. A diagram of the field experimental set up would be useful as an additional panel A. This would allow the reader to clearly differentiate between laboratory and field data. Even though it is described clearly in the text, a graphical representation will help the reader understand the importance of the results here. This is a critical figure that presents crucial data that supports the authors hypothesis, and it needs to be highlighted in the manuscript.

Tables:

No revisions.

Overall, the results are compelling. The manuscript just needs to present the results more clearly.

PLOS authors have the option to publish the peer review history of their article (what does this mean?). If published, this will include your full peer review and any attached files.

Reviewer #1: No

Reviewer #2: No

Figure Files:

Data Requirements:

Reproducibility:

References

---

## [Decision Letter · Decision Letter 2]

13 May 2021

Dear Dr. Wachira,

We are pleased to inform you that your manuscript 'Characterization of a composite with enhanced attraction to savannah tsetse flies from constituents or analogues of tsetse refractory waterbuck (Kobus defassa) body odor' has been provisionally accepted for publication in PLOS Neglected Tropical Diseases.

Best regards,

Rhoel Ramos Dinglasan

Associate Editor

Alvaro Acosta-Serrano

Deputy Editor

Reviewer's Responses to Questions

**Key Review Criteria Required for Acceptance?**

**Methods**

-Are the objectives of the study clearly articulated with a clear testable hypothesis stated?

-Is the study design appropriate to address the stated objectives?

-Is the population clearly described and appropriate for the hypothesis being tested?

-Is the sample size sufficient to ensure adequate power to address the hypothesis being tested?

-Were correct statistical analysis used to support conclusions?

-Are there concerns about ethical or regulatory requirements being met?

Reviewer #2: Yes

**Results**

-Does the analysis presented match the analysis plan?

-Are the results clearly and completely presented?

-Are the figures (Tables, Images) of sufficient quality for clarity?

Reviewer #2: Yes

**Conclusions**

-Are the conclusions supported by the data presented?

-Are the limitations of analysis clearly described?

-Do the authors discuss how these data can be helpful to advance our understanding of the topic under study?

-Is public health relevance addressed?

Reviewer #2: Yes

**Editorial and Data Presentation Modifications?**

Reviewer #2: n/a

**Summary and General Comments**

Reviewer #2: The authors have made enough improvements to the manuscript to warrant publication.

PLOS authors have the option to publish the peer review history of their article (what does this mean?). If published, this will include your full peer review and any attached files.

Reviewer #2: No

---

## [Editor Report · Acceptance letter]

28 May 2021

Dear Dr. Wachira,

We are delighted to inform you that your manuscript, "Characterization of a composite with enhanced attraction to savannah tsetse flies from constituents or analogues of tsetse refractory waterbuck (Kobus defassa) body odor," has been formally accepted for publication in PLOS Neglected Tropical Diseases.

Best regards,

Shaden Kamhawi

co-Editor-in-Chief

Paul Brindley

co-Editor-in-Chief
